# A method for the direct measurement of surface tension of collected atmospherically relevant aerosol particles using atomic force microscopy

**A. D. Hritz, T. M. Raymond, and D. D. Dutcher**

{Bucknell University, Lewisburg, Pennsylvania}

Correspondence to: D. D. Dutcher (dabrina.dutcher@bucknell.edu)

**Abstract**
Accurate estimates of particle surface tension are required for models concerning atmospheric
aerosol nucleation and activation. However, it is difficult to collect the volumes of atmospheric
aerosol required by typical instruments that measure surface tension, such as goniometers or
Wilhelmy plates. In this work, a method that measures, *ex-situ,* the surface tension of collected
liquid nanoparticles using atomic force microscopy is presented. A film of particles is collected
via impaction and is probed using nanoneedle tips with the atomic force microscope. This
micro-Wilhelmy method allows for direct measurements of the surface tension of small
amounts of sample.
This method was verified using liquids whose surface tensions were known. Particles of ozone
oxidized α-pinene, a well-characterized system, were then produced, collected, and analyzed
using this method to demonstrate its applicability for liquid aerosol samples. It was determined
that oxidized α-pinene particles formed in dry conditions have a surface tension similar to that
of pure α-pinene, and oxidized α-pinene particles formed in more humid conditions have a
surface tension that is significantly higher.
**1  Introduction**
According to the Fifth Assessment Report of the Intergovernmental Panel on Climate Change,
clouds and aerosols contribute the largest uncertainty to understanding changes in climate
(Boucher et al. 2013). Aerosols affect the climate directly by reflecting or absorbing solar
radiation, and indirectly when they form cloud particles (Boucher et al. 2013). A major
difficulty in modeling particle nucleation and aerosol activation lies in determining physical
properties of particles on the nanoscale without precise knowledge about chemical composition.
Recent studies in particle nucleation and cloud droplet activation have used various methods to
estimate particle surface tension, which is a very important parameter in modeling both
processes (Duplissy et al. 2008, Kiss et al. 2005, Laaksonen and McGraw 1996, Moldanova
and Ljungström 2000, Petters et al. 2009, Prisle et al. 2010, Sorjamaa et al. 2004, Wex et al.
2009). Particle nucleation is described by the Kelvin equation (Laaksonen and McGraw 1996),
which requires knowledge about surface tension of the nucleating particle (Laaksonen and
McGraw 1996, Schmelzer et al. 1996). Not surprisingly, direct measurement of the surface
tension of particles near activation state conditions has not been possible. Studies on nucleation

often rely on an assumption about the composition and use compiled values for bulk surface tension including values extrapolated from other phases, estimated or interpolated from similar compounds or simply assume "physically reasonable values" (Daisey and Hopke 1993, Moldanova and Ljungström 2000). Hansen et al. (2015), demonstrated the magnitude of the error that can occur when commonly made assumptions about surface tensions are used in models. A direct method of measuring the surface tensions of particles immediately after nucleation is preferable to these assumptions and would likely reduce the error in particle nucleation models.

Köhler theory is used to predict the properties of activating cloud condensation nuclei (Köhler, 1936). The Köhler equation balances the Kelvin effect with Raoult's Law in order to describe particle activation. The Kohler equation applies equilibrium thermodynamics to describe the process in which water vapor condenses to form liquid droplets.

$$\ln\left(\frac{p_w(D_p)}{p^0}\right) = \frac{4M_w\sigma_w}{RT\rho_w D_p} - \frac{6n_s M_w}{\pi\rho_w D_p^3} \tag{1}$$

where, T=absolute temperature, R=ideal gas constant, $p_w$=droplet water vapor pressure, $p_0$=saturation vapor pressure over a flat surface, $\sigma_w$=droplet surface tension, $\rho_w$=denisty of pure water, $n_s$=moles of solute, $M_w$=molecular weight of water, $D_p$=droplet diameter (Köhler 1936). Thus, similar problems arise in specifying physical properties used in the Kelvin term of the Köhler equation. To date, there has been little consistency between assumptions used for the activated particles' surface tensions. Many researchers (Conant et al. 2002, Huff Hartz et al. 2005, Petters and Kreidenweis 2007, Prenni et al. 2007) have assumed that, at activation, the particles consist mainly of water, so a surface tension of pure water was used. Though this is a reasonable initial assumption, it neglects the depressive effect of organic surfactants on the activating particles' surface tensions (Facchini et al. 1999, Kiss et al. 2005). It is now generally agreed upon that, for most activating particles with these surfactants, the surface tension is reduced by about 10-15% (Asa-Awuku et al. 2010, Engelhart et al. 2008, Facchini et al. 2000, King et al. 2009). Several methods have been used to predict this surface tension reduction. Some researchers have collected particles and diluted them so as to allow for a direct measurement using conventional instruments (Asa-Awuku et al. 2008, Henning et al. 2005, Moore et al. 2008, Schwier et al. 2013). These values were then extrapolated back to the initial concentration by fitting them to a Szyskowski-Langmuir isotherm. Occasionally, surface tensions for the particles have been back-calculated using Köhler Theory Analysis when all other parameters are known or estimated (Asa-Awuku et al. 2010, Engelhart et al. 2008). Others

(Kiss et al. 2005, Raymond and Pandis 2002) have prepared solutions mimicking the bulk
chemical composition of aerosol particles and directly measured their surface tensions.
However, none of these methods directly measures the surface tension of the actual particles in
question.
Yazdanpanah (Yazdanpanah et al. 2008) has developed a method to measure the surface tension
of small (~200 nm in diameter) droplets and films using constant-diameter nanoneedle tips on
the atomic force microscope. In this work, we will show how his method has been adapted to
accurately measure the surface tensions of collected atmospheric aerosols.

## 2   Experimental Methods

### 2.1   Particle Generation

In this project, oxidized α-pinene particles were generated in a 1 m$^3$ polytetrafluoroethylene
(PTFE) smog chamber (Fig. 1). Particles were formed in either "dry" (<5% RH) or "wet" (67%
RH) conditions. To generate the "dry" conditions, the chamber was flushed with clean, dry air
for several hours. Compressed air was cleaned using a TSI 3074B filtered air supply. To
generate the "wet" conditions, clean air was bubbled through water at 2 liters per minute (LPM),
filtered, and sent to the smog chamber. The chamber was flushed with this humid air stream
until a maximum relative humidity was reached. Relative humidity was measured using a
Vaisala HM337 Humidity and Temperature Transmitter.  Neither the water content of the
particles nor the surface tensions of the particles generated under dry and wet conditions are
likely to be directly proportional to the relative humidities (Jonsson et al. 2007).
During experiments, the dry, cleaned air stream was sent into the smog chamber at 2 LPM. This
air stream could be diverted either through a sample port or through an ozone generator
(Poseidon Ozone Generator by Ozotech) in series with a HEPA filter before entering the
chamber. An outlet port from the chamber could be connected either to a scanning mobility
particle sizer (a 3080 TSI Differential Mobility Analyzer in series with the 3775 TSI
Condensation Particle Counter) or a cascade impactor (I-1L Cascade impactor by PIXE).
Experiments were only conducted when the initial particle concentration in the smog chamber
was below 100 particles/cm$^3$, as measured by the scanning mobility particle sizer (SMPS).

At the start of each experiment, ozone was added to the smog chamber. If particle counts in the smog chamber remained low after about five minutes, indicating a chamber free of oxidizable volatile organic compounds, 5 μL of liquid α-pinene (97% pure, Acros Organics) was then injected into a sample port, where it was vaporized and carried into the smog chamber. Ozone and α-pinene were added in a roughly 1:1 molar ratio; the high starting concentrations were necessary so that an adequate particle volume would form for collection later. The resulting oxidized α-pinene particles were allowed to age in the chamber for 90 minutes. The ozone-alpha-pinene system was selected because it is one of the more, if not the most, characterized SOA systems. Speciation and chemical characterization results from similar systems have been reported by various researchers (e.g. Jang and Kamens 1999, Praplan et al. 2015, Tu et al. 2016, Yu et al. 1999).

During the aging process, particle size distribution data was collected with the SMPS. The SMPS sample flow rate was 0.3 LPM and the sheath flow rate used on the Differential Mobility Analyzer (DMA) was 3 LPM. These settings allowed for collection of particle size distribution data over the range of 15 to 660 nm. The low sampling flow rate ensured that the smog chamber operated under positive pressure. The size of the oxidized α-pinene particles followed a log-normal distribution whose center shifted to larger sizes over time. In the period where particles aged, the modal diameter increased from around 120 to 200 nm. The most significant changes in particle size distribution occurred in the first hour after the α-pinene was introduced to the smog chamber. The 90 minute aging period ensured minimal changes in particle size distribution during collection. A schematic of the experimental set-up is shown in Figure 1.

## 2.2   Particle Collection

Ninety minutes after α-pinene was introduced to the smog chamber, the outlet of the chamber was switched to feed to the cascade impactor. The second smallest stage (L2) was used to collect the particles on a cleaned steel disk. The 50% aerodynamic cutoff diameter for this stage at 4 LPM was 40 nm. After 90 minutes a visible particle film had collected on the disk. The particles deposit in a circular region ~6 mm in diameter on the steel disc. The steel disc before and after sample collection was imaged using a ScanAsyst and PeakForce Tapping-mode AFM microscope. These images are displayed in Figure 2a and 2b. These images show that the steel disk is rough. After sampling the surface is smoother indicating that the sample flowed and

filled in the roughness. Two traces, centered vertically and horizontally, from each image are
shown in Figure 2c.

### 2.3 Sample Analysis

A Veeco Multimode V Atomic Force Microscope (AFM) and NaugaNeedle NN-HAR-FM60
probes were used to analyze the particle film collected on the disk. The probes consist of a flat,
flexible cantilever, and a nanoneedle mounted normally to the cantilever at its end. The Ga-Ag
nanoneedles are shaped as cylinders on the order of 100 nm in diameter and 10 μm in length.
A micro-Wilhelmy method developed by Yazdanpanah et al. (2008), described below, was then
used to measure the surface tension of the samples.
The sample was analyzed with the AFM in force mode. In this mode, the AFM's piezoelectric
transducers push the sample film up to and away from the probe with high precision. The
downward force exerted on the probe was recorded by the AFM as a function of its location
relative to the film's surface. A force curve obtained with the AFM is presented in Fig. 2.
In Fig. 2, the curve in blue illustrates the force exerted on the probe as it approaches and touches
the sample surface. The curve in red illustrates the force exerted on the probe as it is pulled
from the sample. If it is assumed that only forces related to the surface tension of the liquid film
are exerted on the probe, then equation 2 holds:

$$F_{probe} = \sigma * L * \cos(\theta)$$

18 (2)

where σ is the surface tension of the sample, L is the wetted perimeter of the tip, and θ is the
contact angle between the fluid and the tip. For a more complete derivation of this equation see
(Yum and Yu 2006).
Because the nanoneedle has a cylindrical geometry, the wetted perimeter, L, is constant during
all force measurements. This can be seen by the near-constant negative force exerted on the
probe when it is initially retracting out of the sample. The increase in the downward force before
the nanoneedle is completely pulled from the sample is attributed to a decrease in the contact
angle. At the point the sample breaks away from the nanoneedle, the contact angle is zero.
When this angle is zero and the needle is smaller than the capillary length (Uddin et al. 2011),
equation 2 holds:

$$\sigma = \frac{F_{probe}}{L}$$

(2)

For this project, equation 2 was used (Padday et al. 1975), using the force reading at the point the nanoneedle broke from the sample. This corresponds to point 5 in Figure 3. The magnitude of the force at the break-away step suggests that the collected sample is liquid rather than a glassy or amorphous solid observed for some oxidized VOC systems.

Several aspects of the AFM system were calibrated daily before the collected α-pinene particles were analyzed, typically during particle collection. Because the AFM directly measures the deflection of the cantilever, a force exerted on the nanoneedle could only be obtained after calibrating the cantilever's deflection and determining its spring constant. In the AFM, a laser is reflected off of the cantilever into a photodetector; cantilever deflection is measured by the movement of the laser on the photodetector. To calibrate this measurement, the probe was gently pushed into a hard, steel surface. The slope of the force curve when the probe is in contact with the surface indicates the observed cantilever deflection from the photodetector (y-axis of the force curve) versus the actual distance the surface is moving the cantilever (x-axis of the force curve). This slope was entered into the AFM's operating program.

The spring constant of the tip was found using a thermal tune. The thermal tune is a common method to calculate spring constant using measurements of the cantilever's response to thermal noise (Serry 2005). The native Veeco software was used to perform the thermal tune. After these calibrations, the AFM will produce force curves that relate force and distance accurately.

In order to calculate surface tension from force data, the wetted perimeter of the nanoneedle also had to be obtained. This was done by obtaining force curves of liquid standards and using equation 2 to back-calculate the wetted perimeter given force and surface tension information. Two liquid standards were used: 90% pure oleic acid (Sigma-Aldrich) and 97% pure, non-oxidized liquid α-pinene. The surface tensions of these two standards were measured using a Wilhelmy plate (Sigma 703D, KSV Instruments Ltd.); results are shown in Table 1. Measurements for the standards yielded lower values compared to the literature for pure oleic acid and α-pinene. Because the standards were not completely pure, this was not unexpected, and surface tension values obtained from the Wilhelmy plate were used.

A summary of the steps used to calibrate and analyze samples on the AFM is shown in Figure 4.

## 3    Results and Discussion

Surface tension data was obtained for oxidized α-pinene particles. The AFM's measurements and calculated values are presented in Table 2. Both"dry" oxidized α-pinene particles and "wet" oxidized α-pinene particles were analyzed. The mean surface tension of "dry" oxidized α-pinene particles was found to be 27.5 dyn cm$^{-1}$ at 23 degrees C, with an average uncertainty of 1.1 dyn cm$^{-1}$. This is similar to the surface tension of pure α-pinene as reported in the literature (Daisey and Hopke 1993) and measured with our Wilhelmy plate. The mean surface tension of "wet" oxidized α-pinene particles was found to be 44.4 dyn cm$^{-1}$ at 23 degrees C, with an uncertainty of 2.4 dyn cm$^{-1}$.

The results presented in Table 2 include a set of standards which were done for every set of measurements. The purpose of this standard was to allow the determination of the perimeter of the nano-needle. For the first set of reported measurements, a check standard was also added to verify that the perimeter measurement was correct. This check standard is not required for each set of measurements. The surface tensions measured here were compared to the surface tensions of the standards measured and presented in Table 1.

Table 3 compares the mean surface tensions of oxidized α-pinene particles measured in this study with published estimates for the surface tension of activating, oxidized α-pinene particles (Engelhart et al. 2008, Huff Hartz et al. 2005, Prenni et al. 2007). Our results suggest that the surface tension of dry oxidized α-pinene particles is not very different from the surface tension of its VOC precursor. It is also apparent that the surface tension of oxidized α-pinene particles formed in more humid conditions had a higher surface tension than oxidized α-pinene particles formed in dry conditions.

These results appear to be in agreement with current theory. It is generally believed that the surface tension of an activating oxidized α-pinene particle is slightly lower than that of pure water, at 61.7 dyn cm$^{-1}$ (Engelhart et al. 2008). This is due to the depressive effect of organic surfactants in the droplets. The results from the particles generated at the higher of two humidities suggest that the surface tension is between the surface tension of pure water and the surface tension of the dry oxidized α-pinene. This relationship is unlikely to be directly linear

given that additive surface tensions only apply to chemicals with similar properties which water and the organics produced from the oxidation of alpha-pinene are not. Furthermore, the surface tension of the dry oxidized α-pinene particles was found to be similar to the surface tension of pure α-pinene. This similarity in properties may be due to their similar structures. Now that a method suitable for the direct measurement of particle surface tension has been established, direct measurements of particles with several other moisture contents should be taken to examine the precise relationship between surface tension and moisture content in a particle. With modifications to the particle generation technique, this method can be used to experimentally measure the surface tension of activating particles.

## 4   Conclusions

A method was developed to measure the surface tension of collected liquid aerosol particles using atomic force microscopy. Particles are impacted on a clean surface until a film is formed, then probed with a clean tip in an atomic force microscope. This method minimizes processing of the particles and therefore reduces the risk of sample contamination. The method was verified and calibrated using standard liquids whose surface tensions were in the range of the sample specimens. The standard liquid surface tensions were checked with a Wilhelmy plate. This method does not measure single particle surface tension but does dramatically reduce the required amount of material required to make bulk measurements.

Relatively dry, oxidized α-pinene particles were found to have a surface tension similar to that of pure liquid α-pinene. Oxidized α-pinene particles with higher moisture content were found to have a surface tension significantly higher than that of pure α-pinene, but lower than current assumptions for the surface tension of activating oxidized α-pinene particles. These preliminary results are consistent with the assumption of surface tension depression currently used to approximate the surface tension of activating aerosol particles. With modifications to the particle generation technique, this method can be used to experimentally measure the surface tension of particles closer to activation conditions.

## Acknowledgements

The authors thank NaugaNeedles and Dr. Mehdi Yazdanpanah for supplying the tips for the atomic force microscope and assisting in developing the methods described in this paper. The

1    authors also thank Drs. James Maneval and Ray Dagastine for their assistance.  The authors

2    also appreciate the efforts of Rileigh Casebolt for some of the imaging.

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

Table 1. Surface tension of bulk liquids used for standardization, measured by the Wilhelmy plate at 23.9 degrees C.  Averages reported as "average +/- standard error ($s/\sqrt{n}$)."  Pure oleic acid has a surface tension of 32.79 dyn/cm at 20 degrees C (Chumpitaz et al., 1999), and pure α-pinene has a surface tension of 26.0 dyn/cm at 25 degrees C (Daisey and Hopkey, 1993). Measured values on Wilhelmy plate are close to reported values, considering differences in purity and temperature.

| Component | Surface Tension (dyn/cm) |
|---|---|
| Oleic Acid (90% purity) | 29.47 |
|  | 29.53 |
| Average | 29.50±0.03 |
| α-pinene (97% purity) | 25.75 |
|  | 25.36 |
| Average | 25.6±0.2 |

Table 2. Measured and calculated values obtained during three experiments. In the first
experiment, α-pinene was used as the standard, oleic acid was used as a check standard, and the
oxidized α-pinene particles were generated in dry conditions. In the second experiment, oleic
acid was used as the standard, there was no check standard, and the oxidized α-pinene particles
were generated in dry conditions. In the third experiment, α-pinene was used as the standard,
there was no check standard, and the oxidized α-pinene particles were generated in wet
conditions.

| Experimental conditions | Standard | | Check Standard | | Sample Oxidized α-pinene particles | |
|---|---|---|---|---|---|---|
| | Measured Maximum Force(nN) | Calculated Wetted Tip Perimeter (nm) | Measured Maximum Force(nN) | Calculated Wetted Tip Perimeter (nm) | Measured Maximum Force(nN) | Calculated Surface Tension (dyn/cm) |
| -Particles generated at <10% RH -Standard: α-pinene (97% purity) -Check Standard: Oleic Acid (90% purity) | 9.7 9.6 9.7 | 377.0 373.8 377.0 | 11.0 10.9 10.8 | 373.8 370.7 364.4 | 10.1 10.3 10.2 11.4 10.5 | 26.8 27.4 27.0 30.2 27.8 |
| **Average** | | **375.9±1.1** | | **369.7±2.8** | | **27.8±0.6** |
| -Particles generated at <10% RH -Standard: Oleic Acid (90% purity) -Check Standard: None | 11.7 11.8 11.7 | 395.8 399.0 395.8 | | | 10.8 10.6 10.6 | 27.2 26.7 26.8 |
| **Average** | | **369.9±1.0** | | | | **26.9±0.2** |
| -Particles generated at 67% RH -Standard: α-pinene (97% purity) -Check Standard: None | 12.7 12.5 12.6 | 496.4 490.1 493.2 | | | 21.4 21.2 20.6 22.8 23.4 | 43.3 42.9 41.8 46.3 47.5 |
| **Average** | | **493.2±1.8** | | | | **44.4±1.1** |

1 Table 3. Measured and approximated surface tensions of α-pinene particles. Bulk α-pinene and dry,

2 oxidized α-pinene particles have a similar surface tension. Wet α-pinene particles have a higher surface

3 tension. Measurements from this study are shown in italicized font, other values are given for context.

| RH at particle creation (%) | Surface tension (dyn/cm) | Description, Source |
|---|---|---|
| n/a | **25.6** | Pure α-pinene, bulk<br>This experiment; Wilhelmy plate |
| *<10* | ***27.5*** | *Oxidized α-pinene particles*<br>*This experiment; AFM measurements* |
| *67* | ***44.4*** | *Oxidized α-pinene particles*<br>*This experiment; AFM measurements* |
| >100 (Activation) | **61.7** | Oxidized α-pinene particles, assume depressed surface tension of pure water<br>Engelhart et al., 2008 |
| >100 (Activation) | **72.5** | Oxidized α-pinene particles, assume surface tension of pure water<br>Huff Hartz et al., 2005; Prenni et al., 2007 |

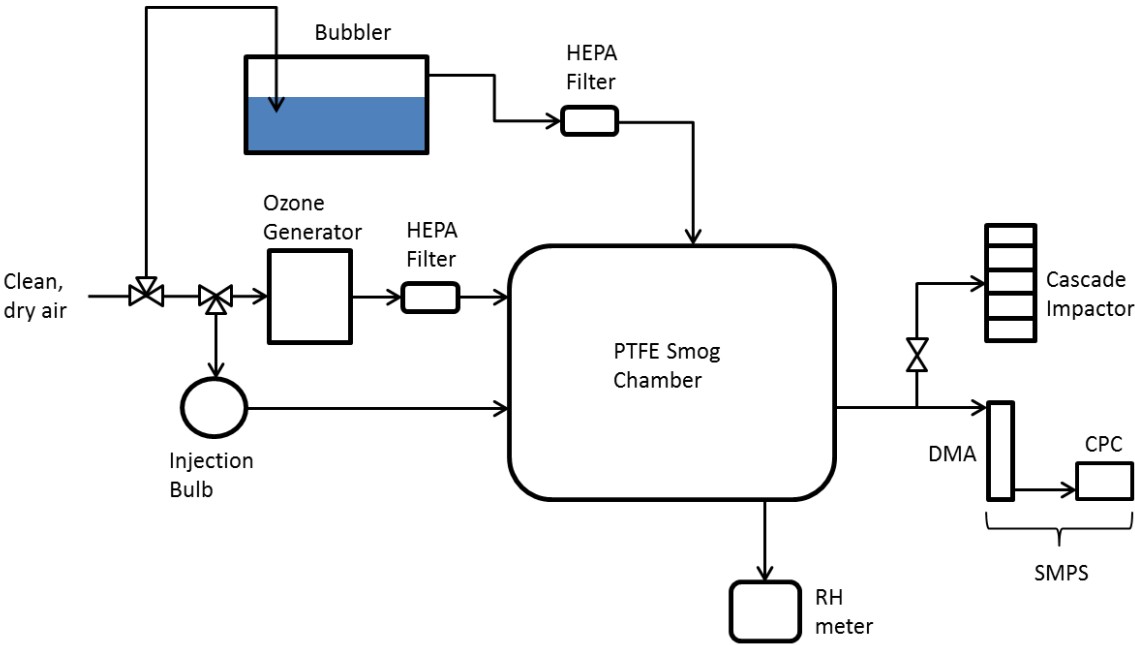

Figure 1. Experimental setup used to generate and collect oxidized α-pinene particles. The smog
chamber is initially flushed with dry or wet air. Once the relative humidity in the chamber is
established, particles are generated in the smog chamber by mixing α-pinene and ozone.
Resulting particles are either analyzed with the SMPS or sampled using the cascade impactor.

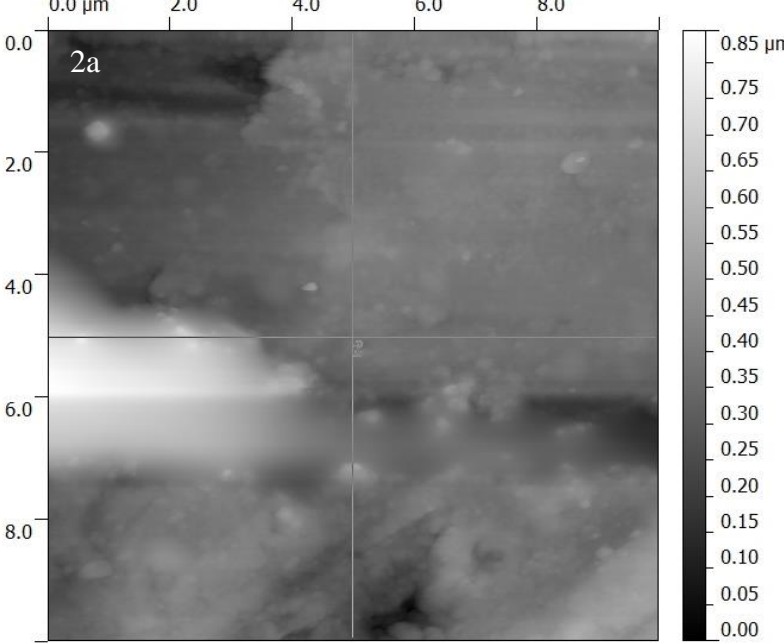

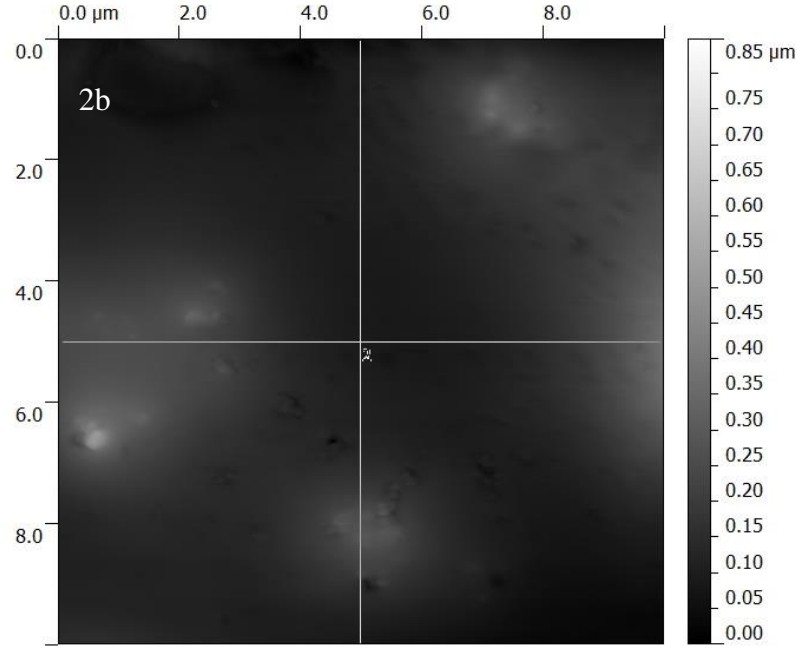

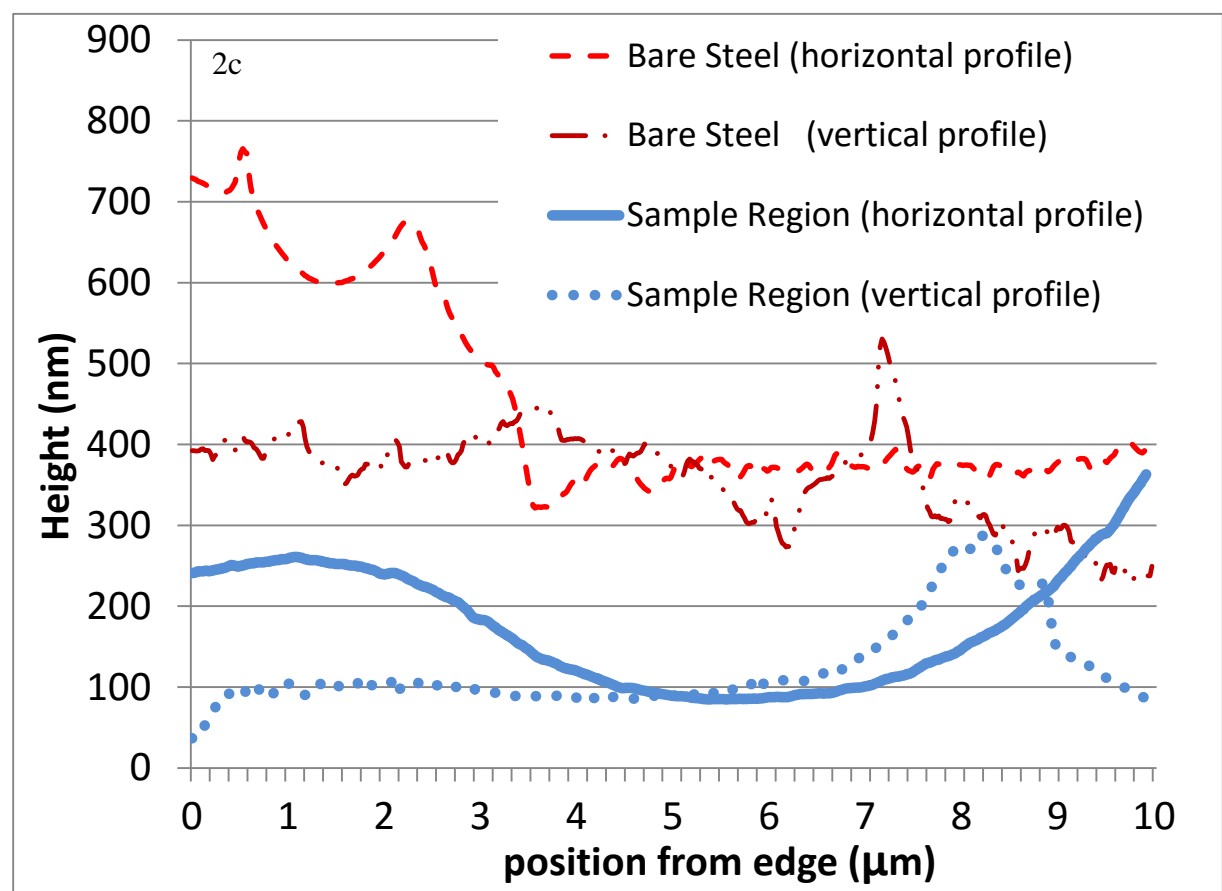

Figures 2a-c. Figure 2a shows an AFM surface scan of cleaned (no sample) puck. The surface

is rough at this microscopic level. Figure 2b shows an AFM scan of the same puck after

sampling. This image was collected in the sample deposit region. It shows that the roughness
gets filled in by the sample. Figure 2c shows the centered vertical and horizontal traces from
these analyses. Significant roughness is observed on the steel that is not observed on the
collected sample. This indicates that the sample could flow, i.e. had liquid characteristics at the
time of sampling.

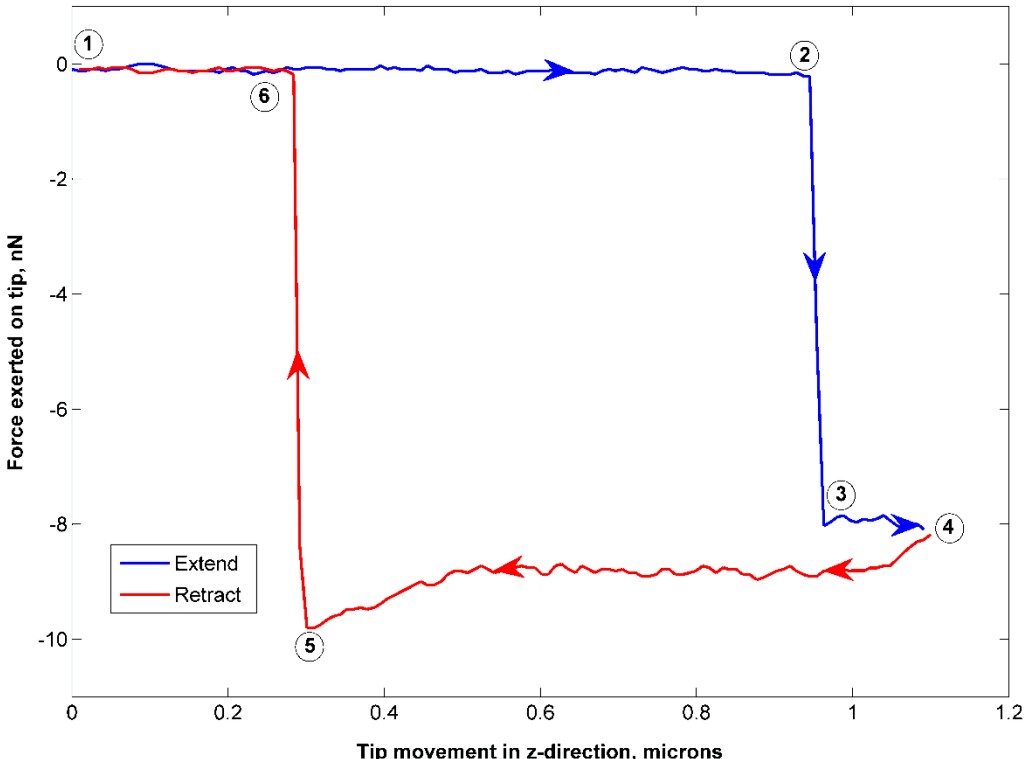

Figure 3. A typical force curve obtained using NaugaNeedle NN-HAR-FM60 probes and an atomic force microscope. The blue line indicates the probe approaching the sample, and the red line indicates the probe retracting from the sample. At point 1, the nanoneedle is approximately 1 micron from the surface of the liquid sample. At point 2, the nanoneedle is just above the surface of the liquid. At point 3, the nanoneedle has touched the liquid, which wicks up and exerts a downward force on the probe. At point 4, the nanoneedle begins to pull out of the liquid. At point 5, the liquid is just about to break from the end of the nanoneedle, and the contact angle of the liquid-needle interface approaches zero. At point 6, the nanoneedle has pulled out of the liquid sample. The probe retracts back to point 1.

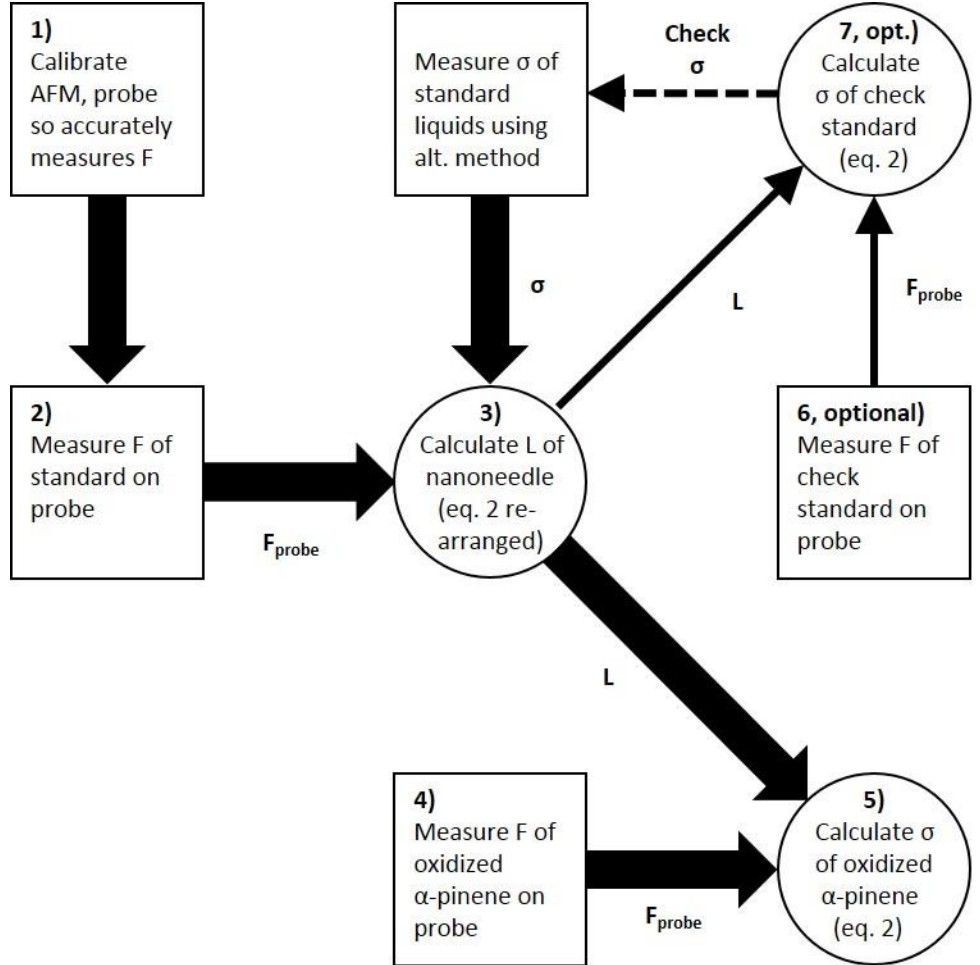

Figure 4. Procedure used to determine the surface tension of oxidized α-pinene particles using
the AFM. The cantilever's spring constant was determined (step 1), which allowed the AFM to
obtain force curves. Force curves of a liquid standard were obtained (step 2), and the
nanoneedle's wetted perimeter was calculated with equation 2 given the standard's known
surface tension (step 3). Force curves of the oxidized α -pinene sample were obtained (step 4),
and its surface tension was calculated with equation 2 given the nanoneedle's wetted perimeter
(step 5). For initial tests, a check standard was used to verify the validity of the wetted perimeter
and sample surface tension calculations (optional steps 6-7).b

