# Peer review of "A method for the direct measurement of surface tension of"

_Atmospheric Chemistry and Physics, 2016_

## Referee Comment (RC1) · Anonymous Referee #1 · 7 Mar 2016

The manuscript "A method for the direct 1 measurement of surface tension of 2 atmospherically relevant aerosol particles using atomic force microscopy" by A. D. Hritz et al. presents a new method for measurement of surface tension for collected liquid nanoparticles using atomic force microscopy. Particle surface tension is a crucial parameter in aerosol thermodynamics and cloud microphysics, but has so far remained an elusive property, due to the lack of experimental methods for single nano-particle surface tension measurements, and the challenges involved in capturing the composition dependent thermodynamic properties of complex chemical mixtures representative of atmospheric aerosols. The method presented here represents one important step towards this goal, enabling direct measurements on much smaller amounts of sample

material, and thus readily applicable to a much wider range of aerosol samples from both laboratory and field measurements than was previously possible.

The manuscript is well-written in clear language with a logic structure and a refreshingly compact presentation. I recommend publishing after addressing three major points, and a few minor issues, as detailed below.

Major points:

1) The authors state in the abstract that the method applies to liquid nano-particles. I am missing a discussion of the actual phase-state of the samples. Are they really all liquid? What are the experimental conditions? It was shown by Virtanen et al. (Nature 467, 824-7, 2010) that chamber generated aerosol of very similar compositions to those sampled here are solid.

2) It is crucially important in the context of nano-particle surface tension to distinguish between the actual surface tension of a single particle and that of a bulk sample comprising a collection of many nano-particles. What are the dimensions of the sampled systems in the presented set-up, i.e. how thick and wide are the collected films on the impactor? Even if one dimension is in the nano-meter range, unless all of them are, chances are that the surface tension will still correspond to that of a much larger macroscopic sample – as the results present here indeed show. Is there any effect of the substrate onto which the sample is collected? Would there be, for certain film thicknesses?

The reason for the importance of this distinction is that interactions between surface area/bulk volume ratio and surfactant bulk/surface partitioning may change the surface tension in aqueous surfactant solutions depending on solution dimensions. This has been predicted from aerosol process models based on Gibbs adsorption theory (see e.g. Prisle et al., Atmos. Chem. Phys., 10, 5663–5683, 2010 and Prisle et al., Atmos. Chem. Phys., 11, 4073–4083, 2011), but not verified by direct measurement, as proper experimental methods are still lacking. It is still of great interest to know the surface

tension of systems with representative chemical compositions, however, as made feasible with the method presented here, since even such information is also still sorely lacking for proper cloud micro-physic modelling.

3) A major concern is that for the results to be applicable in a wider context, they must be accompanied by more detailed or specific chemical information. Surface tension, especially for aqueous surfactants, is a highly non-linear function of composition, for one part due to the non-isotropic nature of the solutions inherent from the surface activity of the organics. Although I understand that such chemical characterization may be beyond the scope of this work, the authors could more carefully distinguish the different systems they are studying and comparing.

First, actual organic composition differences between a-pinene and its oxidation products may indeed cause differences in surface tension of the purely organic phases, and it is indeed plausible that the more oxidized mixtures could have higher surface tensions due to more polar intermolecular interactions. But without any chemical information to facilitate comparison to thermodynamic modelling, or corresponding measurements with conventional methods on samples comprising similar mixtures of oxidation products, it is difficult to arrive at a firm validation of the method presented.

Second, the high-RH oxidation products are likely different than those formed at dry conditions (e.g. Jonsson et al., J. Aerosol Sci., 38, 843– 852, 2007), but again the authors do not comment on the chemical composition or perform parallel conventional measurements, so it is difficult to evaluate the source of the change in surface tension. I would furthermore strongly doubt that the WATER content of these organic mixtures conditioned at ∼65% RH is more than a minor fraction of the particle mass and therefore these particles are essentially highly dilute water solutions in organic solvent. The properties of such mixtures are not immediately comparable to those of dilute aqueous solutions of even the exact same organics (see e.g. a conceptual mixing diagram in Prisle et al., 37, L01802, 2010). This is again due to the highly non-linear variation of thermodynamic properties with composition, for mixtures of molecules with different

structures, such as water and a-pinene and its oxidation products. Specifically, additive surface tensions generally apply most readily for mixtures of components with very similar molecular structures, unlike those where oxidized functionalities or water molecules are introduced to organics. The dilute aqueous organic solutions are representative of the a-pinene oxidation products at the point of CCN activation, as referenced with e.g. Engelhart et al. (Atmos. Chem. Phys., 8, 3937–3949, 2008). I therefore suggest a revision of the discussion, in particular in the paragraph p. 8, l. 3-15, where these differences in sample composition and the implications for the thermodynamic origins of surface tension is taken more closely into account.

Minor comments:

p. 2, l. 26-27: It may be useful to add references to the methods using detailed Gibbs adsorption thermodynamics for evaluating surface tension during cloud activation, e.g. Sorjamaa et al. Atmos. Chem. Phys., 4, 2107–2117, 2004 and previously mentioned Prisle et al., Atmos. Chem. Phys., 10, 5663–5683, 2010.

p. 3, l. 14-16: In connection to the point above, there are a number of studies showing that particles containing strong organic surfactants display cloud activation properties consistent with the surface tension of pure water. This of course does not prove that the droplet surface tensions are in fact identical to that of pure water. Still, there is a convention of using the (counterintuitive) surface tension value of pure water for applications in cloud modelling. Furthermore, several process-level studies have firmly demonstrated that using bulk surface tension values can lead to great over-estimations of particle CCN activity, recently e.g. Hansen et al. Atmos. Chem. Phys., 15, 14071-14089, 2015. These points do not imply that direct measurements of droplet surface tension is not of great interest, on the contrary, it will be a most valuable addition to the field to be able to determine the actual in situ droplet surface tension. As the authors note, none of the existing methods "directly measures the surface tension of the actual particles in question" (p. 3, l. 25-26). Furthermore, both the methods calculating surface tensions from Gibbs adsorption theory, and those applying back-calculation

from observed cloud activation, rely on assumptions of droplet properties in addition to surface tension, which have also not been verified by direct and specific measurement.

p. 7, l. 29: The particles measured by e.g. Engelhart are also the a-pinene oxidation products.

Table 1: How is the "standard error" defined? Is it e.g. taken as the standard deviation or twice that quantity? What temperature were the present measurements made at?

Table 2: I think it would be useful to clarify further what the functions of the standard and check standard, respectively, are. Specifically, what is the relationship between the surface tension of these standards (why is it not given in the table directly, or why is there not an explicit reference to where the values are found elsewhere in the manuscript) and the sample surface tensions?

Table 3: The RH is not 100% at activation. These particles require supersaturated conditions for CCN activation. Note that the water content for mixtures conditioned at these different humidities is by no means implicit.

---

## Referee Comment (RC2) · Anonymous Referee #2 · 7 Mar 2016

**General comments**

The manuscript entitled: "A method for the direct measurement of surface tension of atmospherically relevant aerosol particles using atomic force microscopy" describes how to use a micro-Wilhelmy method to measure the surface tension of collected aerosol liquid in an atomic force microscope. While the manuscript is generally well-written, clear, very concise, and contains no major errors that I can tell, I wonder if it is, indeed, appropriate for publication in ACP.

From ACP's website (http://www.atmospheric-chemistry-and-physics.net/about/aims_and_scope.html), "*The journal scope is focused on studies with general implications for atmospheric science rather than investigations that are*

[Figure]

*primarily of local or technical interest*." While there are some general implications from this study, namely, that it presents a new way to measure aerosol liquid surface tension in a direct way that uses much less sample liquid than traditional bulk methods require, the focus of this manuscript seems to be on the technical method development. The more general application of this method was to one aerosol system, oxidized alpha-pinene SOA, under dry and moderately wet conditions. The analysis was extremely short and contained no other supporting information or measurements other than a comparison to a couple prior studies in the literature that don't actually match these results or methods well at all. For better "general implications for atmospheric science," the results section needs to be developed much more extensively.

Perhaps other journals to consider submitting to would be Atmospheric Measurement Techniques or Aerosol Science & Technology. AMT's aims and scopes include "*the development, intercomparison, and validation of measurement instruments and techniques of data processing and information retrieval for gases, aerosols, and clouds.*" AST's aims and scopes include "*instrumentation for the measurement of aerosol physical, optical, chemical and biological properties.*" I am sure there are other appropriate journals available as well.

**Specific comments**

One major consideration the authors might also make is changing the title. The "direct measurement of ... aerosol particles" implies that the method is measuring surface tension on an individual aerosol particle; but this is not true. In reality, lots of aerosol particles are collected on a plate until there is a thin film of liquid covering the plate. In essence, this is still a *bulk* technique for measuring surface tension, just that this *bulk* is much, much smaller than traditional bulk methods (e.g. the traditional Wilhelmy plate method).

pg 2 - line 11 : Are the results presented in this manuscript really "preliminary"? What will it take to make them final? Only final results (i.e. calibrated measurements from working instruments) should be published.

2 - 28 : Because this is a methods paper, it would be helpful to include equations and define each term and give the implications of each term of the equation. These include the Kelvin equation, Kohler equation (pg 3, line 6) and Raoult's Law (pg 3, line 6).

3 - 1 : What do you mean by "tabulated values"? Also, the literature review on surface tension measurements should be expanded - certainly there are more than two papers that talk about surface tension values with regards to nucleation?

3 - 3 : Is "shortly after nucleation" referring to a time issue or a particle size issue?

3 - 12 : What does "good" mean? What basis do you have for making the "good initial assumption" claim? This phrase is confusing, especially given the next sentence which says the surface tension is reduced by 10-15%.

4 - 6, 7 : Define "clean" air. What filters/scrubbers do you use? Is the air from a cylinder or compressed air line?

4 - 25-26 : This is the same basic sentence as pg 5, lines 4-5.

5 - 20 : To be clear, are you paraphrasing this method in the next few paragraphs? You should make it clear that the reader does or does not need to read the Yazdanpanah et al., 2008 paper.

5 - 29 : Is there a citation for equation 1?

6 - 3 : Is there an image that can be used to demonstrate the contact angle? Perhaps this can be included in a supplemental section.

6 - 10 : Explain better what capillary length means in the context of this measurement technique. Perhaps include an equation. This explanation is important to justify use of equation 2 as the basis for the technique.

6 - 23 : I do not understand this calibration method. It sounds like you are pushing the needle into a hard surface and deflecting the needle. But it must be the case that the needle actually stays rigid and the deflection comes where the needle is connected to something else that measures force. Perhaps a figure describing this process with some actual images would be helpful. Also, if what I have described is how the calibration is done, then how do you relate deflection to force?

7 - 3 : What is a "thermal tune" and how exactly is it done?

7 - 15 : There is no mention of a "check standard" until later, which can be confusing when first looking at Figure 3. Also, where is the hard, steel surface used for calibration?

8 - 13-14 : What modifications? Explain in much greater detail what you mean by this sentence.

8 - 32 : Use of "particles at activation" implies measurement of individual small droplets - this is not true with these current methods. Can these "modification" (list them) actually get to measuring surface tension of individual droplets?

Table 1 : The term "reasonably close" is subjective and should be avoided. What temperature were your measurements taken at? Did you only take two measurements per solution? I think measurements should usually be done at least three times and then averaged. For these compounds, are there any temperature-dependent models of surface tension that you could use to better compare to your measurements at whatever temperature your measurements were taken at?

Table 3 : Which measurements are approximated versus measured? I can make a guess, but this should be much clearer in the table. In the caption, the last two sentences look like analysis, which should go in the body of the manuscript rather than in the caption. Where does the surface tension value of 27.5 come from? I can't find it on Table 2 anywhere. Also, Table 3 should have uncertainties next to each surface

tension value.

**Technical corrections**

A careful proofread of the entire manuscript should be done – there are several places where no space exists between words and several places where multiple spaces exist between words.

pg 2 - line 17 : Do not need this line; this information is in the citation.

2 - 30 : What is "this type"? Be more clear and specific.

3 - 23 : I am not sure ACP's convention, but should references be moved to the end of the sentence? This isn't the only sentence in the manuscript written this way, but this one first caught my attention as one where the reference could be moved to the end without sacrificing any meaning.

6 - 26-27 : This is an unnecessary detail.

7 - 22 : Fix "a an average"

8 - 10 : What is "This"?

---

## Author Comment (AC1) · 12 May 2016

Please see files in attached zip folder.

Please also note the supplement to this comment:
http://www.atmos-chem-phys-discuss.net/acp-2016-35/acp-2016-35-AC1-supplement.zip

---

## Author Response (AR1)

**Authors' Response to the Review Comments**

*Journal:*          Atmos. Chem. Phy.acp-2016-35

*Manuscript #:*     acp-2016-35

*Title of Paper:*   A Method for the Direct Measurement of surface tension of atmospherically relevant aerosol particles using Atomic Force Microscopy

*Authors:*          A.D. Hritz, T.M. Raymond and D.D. Dutcher

*Date Sent:*        5/11/2016

We appreciate the time and efforts of the editors and both referees in reviewing this manuscript. We have addressed the issues indicated in the reviewer's reports, and believe that the revised version meets the journal publication requirements. The reviewer's efforts have led to a clearer and better manuscript.

**Response to Comments from Reviewer 1**

Reviewer's comments are italicized while author responses are not. Page and line locations refer to the Markup version.

**Reviewer's Comment with Responses:**

*The manuscript "A method for the direct 1 measurement of surface tension of 2 atmospherically relevant aerosol particles using atomic force microscopy" by A. D. Hritz et al. presents a new method for measurement of surface tension for collected liquid nanoparticles using atomic force microscopy. Particle surface tension is a crucial parameter in aerosol thermodynamics and cloud microphysics, but has so far remained an elusive property, due to the lack of experimental methods for single nano-particle surface tension measurements, and the challenges involved in capturing the composition dependent thermodynamic properties of complex chemical mixtures representative of atmospheric aerosols. The method presented here represents one important step towards this goal, enabling direct measurements on much smaller amounts of sample material, and thus readily applicable to a much wider range of aerosol samples from both laboratory and field measurements than was previously possible.*

*The manuscript is well-written in clear language with a logic structure and a refreshingly compact presentation. I recommend publishing after addressing three major points, and a few minor issues, as detailed below.*

*Major points: 1) The authors state in the abstract that the method applies to liquid nano-particles. I am missing a discussion of the actual phase-state of the samples. Are they really all liquid? What are the experimental conditions? It was shown by Virtanen et al. (Nature 467, 824-7, 2010) that chamber generated aerosol of very similar compositions to those sampled here are solid.*

These measurements would not be possible to make on non-liquid samples.   If the samples were not liquid, they could not wick up onto the nano-needle tip.   To confirm that the samples were liquid upon collection as well as analysis, additional AFM measurements were made.   These have been added to the paper. (p 5 1-6, p 7 12-14 and Figure 2).   These measurements do indicate that the samples are liquid.   To summarize, AFM images were obtained of one of the stainless steel discs before and after sampling.   The image collected before sample collection shows an irregular, rough surface.   The image collected after sample collection shows a smooth surface.   During sample collection, the sample flowed to fill in the rough surface and leave a smooth surface.

The paper referenced by the reviewer (Virtanen et al) used a similar method of oxidizing VOCs but their VOCS were a much more complex mixture of biogenic VOCs which may be what leads to the difference in sample phase.

*2) It is crucially important in the context of nano-particle surface tension to distinguish between the actual surface tension of a single particle and that of a bulk sample comprising a collection of many nano-particles. What are the dimensions of the sampled systems in the presented set-up, i.e. how thick and wide are the collected films on the impactor? Even if one dimension is in the nano-meter range, unless all of them are, chances are that the surface tension will still correspond to that of a much larger macroscopic sample – as the results present here indeed show. Is there any effect of the substrate onto which the sample is collected? Would there be, for certain film thicknesses? The reason for the importance of this distinction is that interactions between surface area/bulk volume ratio and surfactant*

*bulk/surface partitioning may change the surface tension in aqueous surfactant solutions depending on solution dimensions. This has been predicted from aerosol process models based on Gibbs adsorption theory (see e.g. Prisle et al., Atmos. Chem. Phys., 10, 5663–5683, 2010 and Prisle et al., Atmos. Chem. Phys., 11, 4073–4083, 2011), but not verified by direct measurement, as proper experimental methods are still lacking. It is still of great interest to know the surface tension of systems with representative chemical compositions, however, as made feasible with the method presented here, since even such information is also still sorely lacking for proper cloud micro-physic modelling.*

Our method is NOT a single particle method.   We have added additional language (Title, p 2 5, p 5 1-6, p 9 28-29 ) trying to ensure that there is no confusion about this issue.   Although the exact dimensions of the deposit are not known, it is certain that the width of the sample is above the nanometer range. The sample image added to confirm the sample phase (Figure 2b) shows that the sample width is at least 10 micrometers.   We concur with the reviewer's point that this is likely to result in measurements that are closer to bulk measurements than single particle measurement.   The purpose and strength of this method is to reduce the quantity of sample required to make these "bulk" measurements.

*3) A major concern is that for the results to be applicable in a wider context, they must be accompanied by more detailed or specific chemical information. Surface tension, especially for aqueous surfactants, is a highly non-linear function of composition, for one part due to the non-isotropic nature of the solutions inherent from the surface activity of the organics. Although I understand that such chemical characterization may be beyond the scope of this work, the authors could more carefully distinguish the different systems they are studying and comparing. First, actual organic composition differences between a-pinene and its oxidation products may indeed cause differences in surface tension of the purely organic phases, and it is indeed plausible that the more oxidized mixtures could have higher surface tensions due to more polar intermolecular interactions. But without any chemical information to facilitate comparison to thermodynamic modelling, or corresponding measurements with conventional methods on samples comprising similar mixtures of oxidation products, it is difficult to arrive at a firm validation of the method presented. Second, the high-RH oxidation products are likely different than those formed at dry conditions (e.g. Jonsson et al., J. Aerosol Sci., 38, 843– 852, 2007), but again the authors do not comment on the chemical composition or perform parallel conventional measurements,*

*so it is difficult to evaluate the source of the change in surface tension. I would furthermore strongly doubt that the WATER content of these organic mixtures conditioned at ∼65% RH is more than a minor fraction of the particle mass and therefore these particles are essentially highly dilute water solutions in organic solvent. The properties of such mixtures are not immediately comparable to those of dilute aqueous solutions of even the exact same organics (see e.g. a conceptual mixing diagram in Prisle et al., 37, L01802, 2010). This is again due to the highly non-linear variation of thermodynamic properties with composition, for mixtures of molecules with different structures, such as water and a-pinene and its oxidation products. Specifically, additive surface tensions generally apply most readily for mixtures of components with very similar molecular structures, unlike those where oxidized functionalities or water molecules are introduced to organics. The dilute aqueous organic solutions are representative of the a-pinene oxidation products at the point of CCN activation, as referenced with e.g. Engelhart et al. (Atmos. Chem. Phys., 8, 3937–3949, 2008). I therefore suggest a revision of the discussion, in particular in the paragraph p. 8, l. 3-15, where these differences in sample composition and the implications for the thermodynamic origins of surface tension is taken more closely into account.*

We agree entirely with the reviewer's comments on this point.   However, we do not believe that we made some of the claims to which the comments would apply.   We have revised the paragraph (p 9 7-12) suggested by the reviewer to clarify this issue. The oxidized alpha-pinene system was chosen as our demonstration system because it is one of the most characterized secondary organic aerosol systems. The purpose of generating the particles at two different humidities was to demonstrate the difference in the response of the measurement method to what the reviewer correctly suggests is only likely to be a minor change in the composition of the particles.   This helps demonstrate a minimal level of sensitivity. We do not claim that the change in humidity during the particle generation is directly proportional to the change in the water content of the particles.   It is our hope that this method can be extended, either by us or by other researchers to investigate the surface tension of collected particles at the point of CCN activation.   However, those measurements are beyond the scope of this current manuscript.

*Minor comments: p. 2, l. 26-27: It may be useful to add references to the methods using detailed Gibbs adsorption thermodynamics for evaluating surface tension during cloud activation, e.g. Sorjamaa et al. Atmos. Chem. Phys., 4, 2107–2117, 2004 and previously mentioned Prisle et al., Atmos. Chem. Phys., 10,*

*5663–5683, 2010.*

Excellent suggestion.    Done!

*p. 3, l. 14-16: In connection to the point above, there are a number of studies showing that particles containing strong organic surfactants display cloud activation properties consistent with the surface tension of pure water. This of course does not prove that the droplet surface tensions are in fact identical to that of pure water. Still, there is a convention of using the (counterintuitive) surface tension value of pure water for applications in cloud modelling. Furthermore, several process-level studies have firmly demonstrated that using bulk surface tension values can lead to great over-estimations of particle CCN activity, recently e.g. Hansen et al. Atmos. Chem. Phys., 15, 14071- 14089, 2015. These points do not imply that direct measurements of droplet surface tension is not of great interest, on the contrary, it will be a most valuable addition to the field to be able to determine the actual in situ droplet surface tension. As the authors note, none of the existing methods "directly measures the surface tension of the actual particles in question" (p. 3, l. 25-26). Furthermore, both the methods calculating surface tensions from Gibbs adsorption theory, and those applying back-calculation from observed cloud activation, rely on assumptions of droplet properties in addition to surface tension, which have also not been verified by direct and specific measurement.*

The Hansen reference has been added to the introduction    as part of the motivation for this work

*p. 7, l. 29: The particles measured by e.g. Engelhart are also the a-pinene oxidation products.*

We have clarified the language in this section to clarify that it is not just alpha pinene but oxidized alpha pinene that was being studied.

*Table 1: How is the "standard error" defined? Is it e.g. taken as the standard deviation or twice that quantity? What temperature were the present measurements made at?*

We used the statistical definition of standard error. It is calculated as the standard deviation of the sampling distribution of a statistic (s*n^-0.5)    The temperature at which the bulk measurements were made (23.9 °C) has also been added (Figure 1)

*Table 2: I think it would be useful to clarify further what the functions of the standard and check standard, respectively, are. Specifically, what is the relationship between the surface tension of these standards (why is it not given in the table directly, or why is there not an explicit reference to where the values are found elsewhere in the manuscript) and the sample surface tensions?*

A paragraph describing the purpose of the check standards and clarifying the data used for comparison (the values presented in Table 1) has been added (p 8 22-27).   This suggestion is quite useful as we had not previously clarified the role of the check standard.

*Table 3: The RH is not 100% at activation. These particles require supersaturated conditions for CCN activation. Note that the water content for mixtures conditioned at these different humidities is by no means implicit.*

The table has been altered to reflect the supersaturation condition required for activation (Table 3).   We do not mean to imply that the water content and humidities are directly proportional. We added the humidity information of our measurements and the others only to give context to our measurements.

**Response to Comments from Reviewer 2**

Reviewer's comments are italicized while author responses are not. Page and line locations refer to the Markup version.

*The manuscript entitled: "A method for the direct measurement of surface tension of atmospherically relevant aerosol particles using atomic force microscopy" describes how to use a micro-Wilhelmy method to measure the surface tension of collected aerosol liquid in an atomic force microscope. While the manuscript is generally well-written, clear, very concise, and contains no major errors that I can tell, I wonder if it is, indeed, appropriate for publication in ACP.*

*From ACP's website (http://www.atmospheric-chemistry-andphysics.net/about/aims_and_scope.html), "The journal scope is focused on studies with general implications for atmospheric science rather than investigations that are primarily of local or technical interest." While there are some general implications from this study, namely, that it presents a new way to measure aerosol liquid surface tension in a direct way that uses much less sample liquid than traditional bulk methods require, the focus of this manuscript seems to be on the technical method development. The more general application of this method was to one aerosol system, oxidized alpha-pinene SOA, under dry and moderately wet conditions. The analysis was extremely short and contained no other supporting information or measurements other than a comparison to a couple prior studies in the literature that don't actually match these results or methods well at all. For better "general implications for atmospheric science," the results section needs to be developed much more extensively.*

*Perhaps other journals to consider submitting to would be Atmospheric Measurement Techniques or Aerosol Science & Technology. AMT's aims and scopes include "the development, intercomparison, and validation of measurement instruments and techniques of data processing and information retrieval for gases, aerosols, and clouds." AST's aims and scopes include "instrumentation for the measurement of aerosol physical, optical, chemical and biological properties." I am sure there are other appropriate journals available as well.*

The authors feel that this is an issue for the editor to determine.   Although Atmos. Chem. Phys. is not primarily a methods journal they are the preeminent journal in the broader topic of aerosol surface tension.   And other paper like ours, method based but relating to larger issues, have regularly been published in this journal.   We would consider a transfer to AMT at the editor's suggestion.

*Specific comments*

*One major consideration the authors might also make is changing the title. The "direct measurement of ... aerosol particles" implies that the method is measuring surface tension on an individual aerosol particle; but this is not true. In reality, lots of aerosol particles are collected on a plate until there is a thin film of liquid covering the plate. In essence, this is still a bulk technique for measuring surface tension, just that this bulk is much, much smaller than traditional bulk methods (e.g. the traditional Wilhelmy plate method).*

We agree with the reviewer on this point, this is not a single particle method and we never intended to imply that it was. We have added additional language (Title, p 2 5, p 5 1-6, p 9 28-29 ) trying to ensure that there is no confusion about this issue and to clarify that the particles were collected before analysis. This should reduce the potential for confusion about this topic.

*pg 2 - line 11 : Are the results presented in this manuscript really "preliminary"? What will it take to make them final? Only final results (i.e. calibrated measurements from working instruments) should be published.*

True! This has been fixed.

*2 - 28 : Because this is a methods paper, it would be helpful to include equations and define each term and give the implications of each term of the equation. These include the Kelvin equation, Kohler equation (pg 3, line 6) and Raoult's Law (pg 3, line 6).*

We added the full Kohler equation to the paper along with a reference that contains the other equations requested with more information about their derivation.

*3 - 1 : What do you mean by "tabulated values"? Also, the literature review on surface tension measurements should be expanded - certainly there are more than two papers that talk about surface tension values with regards to nucleation?*

We have added more detail about how the values cited above were tabulated. We also added additional references that talk about surface tension values with regards to nucleation.

*3 - 3 : Is "shortly after nucleation" referring to a time issue or a particle size issue?*

We have clarified this language.

*3 - 12 : What does "good" mean? What basis do you have for making the "good initial assumption" claim? This phrase is confusing, especially given the next sentence which says the surface tension is reduced by 10-15%.*

The "good" in this case referred to reasonable choice for the initial surface tensions assumptions in system models.   We have changed it to "reasonable" to avoid any confusion.

*4 - 6, 7 : Define "clean" air. What filters/scrubbers do you use? Is the air from a cylinder or compressed air line?*

Details of our compressed air system have been added.

*4 - 25-26 : This is the same basic sentence as pg 5, lines 4-5.*

The sentence with repeated information has been eliminated.

*5 - 20 : To be clear, are you paraphrasing this method in the next few paragraphs? You should make it clear that the reader does or does not need to read the Yazdanpanah et al., 2008 paper.*

We leave this decision to the reader of the paper.   We summarize the most important aspects of Yazdanpanah's paper, especially the parts that apply to our research.   However, we'd never advise against reading the other paper for more detail if a reader so wished.

*5 - 29 : Is there a citation for equation 1?*

A citation that contains the equation along with additional derivation information has been added.

*6 - 3 : Is there an image that can be used to demonstrate the contact angle? Perhaps this can be included in a supplemental section.*

We are not aware of a photograph showing this angle.  (We also can't tell if the reviewer is requesting an actual photograph or just a schematic.  If the schematic is what is being suggested, we can add that but without that clarification we will wait.)

*6 - 10 : Explain better what capillary length means in the context of this measurement technique. Perhaps include an equation. This explanation is important to justify use of equation 2 as the basis for the technique.*

A reference has been added that contains more information about capillary lengths.

*6 - 23 : I do not understand this calibration method. It sounds like you are pushing the needle into a hard surface and deflecting the needle. But it must be the case that the needle actually stays rigid and the deflection comes where the needle is connected to something else that measures force. Perhaps a figure describing this process with some actual images would be helpful. Also, if what I have described is how the calibration is done, then how do you relate deflection to force?*

The reviewer's understanding is correct.  It is not that the needle is deflecting but the cantilever (which is part of the AFM tip) that flexes as the nano-needle is pushed into a hard surface.  The calibration of deflection sensitivity is a common procedure among AFM users.  A reference to the manufacturer of our instrument (with associated manuals) has been added though this reference might not be relevant to users of different brands of AFMs.

*7 - 3 : What is a "thermal tune" and how exactly is it done?*

A thermal tune is used to determine the spring constant of an individual AFM tip.  Each manufacturer of AFMs has implemented this tune somewhat differently. A reference to a white paper on our instrument's manufacturer method has been added.  Similarly to the previous comment, this is a common procedure and a detailed description of how to do it on one instrument is not necessarily pertinent to the use of another instrument.

*7 - 15 : There is no mention of a "check standard" until later, which can be confusing when first looking at Figure 3. Also, where is the hard, steel surface used for calibration?*

This is an excellent point, noted by both reviewers.   We have added information defining the purpose and our use of a check standard in this system.   The thermal tune and deflection calibration are included in figure 4 (the old figure 3) in the first step.   Without these calibrations, the Force (F in step 1) cannot be measured accurately.

*8 - 13-14 : What modifications? Explain in much greater detail what you mean by this sentence.*

We are suggesting a potential important extension of this methodology in this sentence.   Until the experiment has actually been done, we hesitate to suggest exactly how the particle generation and collection methodologies should be modified.   We have removed the word "minor" from the description as that was an assumption on our part the veracity of which has not been confirmed yet.

*8 - 32 : Use of "particles at activation" implies measurement of individual small droplets - this is not true with these current methods. Can these "modification" (list them) actually get to measuring surface tension of individual droplets?*

We are in complete agreement with the reviewer about the limitations of this technique.   It is not a single particle technique and it is unlikely (though perhaps not impossible) that it will ever be modified for single particle measurements.

*Table 1 : The term "reasonably close" is subjective and should be avoided. What temperature were your measurements taken at? Did you only take two measurements per solution? I think measurements should usually be done at least three times and then averaged. For these compounds, are there any temperature-dependent models of surface tension that you could use to better compare to your measurements at whatever temperature your measurements were taken at?*

Unfortunately, the literature values referenced in this table did not all contain information about the purity of the chemicals being tested.   Our measurements were all made at 23.9 °C.   This information has been added to this table.   However, it is not possible to accurately model a complete correction to our measurements without the previously mentioned missing information.   We agree with the reviewer that increasing the number of measurements can decrease the uncertainty associated with a measurement. However, it may not reduce to total error depending on the source of the error.   The two sets of measurements that were made are in statistical agreement (0.2 and 1.5% relative error).   The utility of making additional measurements at this point would be questionable due to the time offset between the original measurements and any potential current measurements.

*Table 3 : Which measurements are approximated versus measured? I can make a guess, but this should be much clearer in the table. In the caption, the last two sentences look like analysis, which should go in the body of the manuscript rather than in the caption. Where does the surface tension value of 27.5 come from? I can't find it on Table 2 anywhere. Also, Table 3 should have uncertainties next to each surface tension value.*

The values from our experiments are all measured.   The values from Engelhart et al, Huff Hartz et al and Prenni et al are all provided to give context to our values.   It demonstrates that our surface tension values for particles generated in dry conditions are similar to the bulk un-oxidized liquid while our surface tension values for the particle generated in wet conditions are more similar to those used and published by the authors cited above.   We have changed the formatting of this table to indicate which value were measured by us in the hope of reducing any potential confusion.

*Technical corrections*
*A careful proofread of the entire manuscript should be done – there are several places where no space exists between words and several places where multiple spaces exist between words.*

We have reviewed our manuscript and hope that we caught and corrected these typographical errors.

*pg 2 - line 17 : Do not need this line; this information is in the citation.*

We do not agree that all potential readers of this manuscript will recognize the reference. So we would prefer to provide the information more directly even if it is technically repetitious.

*2 - 30 : What is "this type"? Be more clear and specific.*

We have clarified the language in this line to be clear that we are discussing the surface tension of particles near activation conditions.    This should be clearer and more specific.

*3 - 23 : I am not sure ACP's convention, but should references be moved to the end of the sentence? This isn't the only sentence in the manuscript written this way, but this one first caught my attention as one where the reference could be moved to the end without sacrificing any meaning.*

Several of the references have been moved for the references that we deemed would not lose meaning with said change.

*6 - 26-27 : This is an unnecessary detail.*

We wish to be clear that the calibration constant (slope) is automatically applied by the instrument operating software and not post-applied in our calculation.

*7 - 22 : Fix "a an average"*

Done!

*8 - 10 : What is "This"?*

Clarified!

[revised manuscript text omitted]
** **Kyungsuk Yum** and **Min-Feng Yu** * Department of Mechanical and Industrial Engineering, University of Illinois at Urbana-Champaign, 1206 West Green Street, Urbana, Illinois 61801 *Nano Lett.*, **2006**, *6* (2), pp 329–333 **DOI:** 10.1021/nl052084l

Because the nanoneedle has a cylindrical geometry, the wetted perimeter, L, is constant during all force measurements. This can be seen by the near-constant negative force exerted on the probe when it is initially retracting out of the sample. The increase in the downward force before the nanoneedle is completely pulled from the sample is attributed to a decrease in the contact angle. At the point the sample breaks away from the nanoneedle, the contact angle is zero. When this angle is zero and the needle is smaller than the capillary length (Uddin et al., 2011), equation 2 holds:

$$\sigma = \frac{F_{probe}}{L}$$

(2)

For this project, equation 2 was used, using the force reading at the point the nanoneedle broke from the sample. This corresponds to point 5 in Figure 23. The magnitude of the force at the break-away step suggests that the collected sample is liquid rather than a glassy or amorphous solid observed for some oxidized VOC systems.

Several aspects of the AFM system were calibrated daily before the collected -pinene particles were analyzed, typically during particle collection. Because the AFM directly measures deflection of the cantilever, a force exerted on the nanoneedle could only be obtained after calibrating the cantilever's deflection and determining its spring constant. In the AFM, a laser is reflected off of the cantilever into a photodetector; cantilever deflection is measured by movement of the laser on the photodetector. To calibrate this measurement, the probe was gently pushed into a hard, steel surface. The slope of the force curve when the probe is in contact with the surface indicates the observed cantilever deflection from the photodetector (y-axis of the force curve) versus the actual distance the surface is moving the cantilever (x-axis of the force curve). This slope was entered into the AFM's operating program.

The spring constant of the tip was found using a thermal tune. The thermal tune is a common method to calculate spring constant using measurements of the cantilever's response to thermal noise (Serry, 2010). The native Veeco software was used to perform the thermal tune. After these calibrations, the AFM will produce force curves that relate force and distance accurately.

Comment [DDD3]: http://pubs.rsc.org/en/Content/ArticleLanding/1975/F1/f19757101919#!divAbstract

Comment [DDD4]: add bruker.com reference

Comment [DDD5]: http://www.bruker.co.jp/axs/nano/imgs/pdf/AN090.pdf

[revised manuscript text omitted]